# Clinical Nurses’ Involvement in Decision-Making Process at the Nursing Unit-Based Council Level: A Cross-Sectional Study of Shared Professional Governance in the Kingdom of Saudi Arabia

**DOI:** 10.3390/nursrep15120426

**Published:** 2025-11-28

**Authors:** Regie Buenafe Tumala

**Affiliations:** Medical-Surgical Department, College of Nursing, King Saud University, Riyadh 12372, Saudi Arabia; rtumala@ksu.edu.sa

**Keywords:** clinical nurse, decision-making, involvement, registered nurse, nursing governance, shared governance, unit-based council

## Abstract

**Background:** The implementation of shared governance within the nursing practice results in heightened satisfaction among nurses and enhances the quality of care provided. Shared governance fosters collaborative and proactive relationships among nurses and healthcare providers, while also cultivating a sense of confidence among nurses. However, evidence suggests that a lack of awareness, reliance on traditional governance, and inadequate shared governance structures among nurses continue to exist at the unit-based council (UBC) level, including those in the Kingdom of Saudi Arabia (KSA). **Purpose:** The present study aimed to assess the extent of clinical nurses’ perceptions concerning shared governance at the UBC level, and to examine the variations and relationships in their perceptions based on demographic and work-related characteristics. It further explored the demographic and work-related factors that affect the overall perceptions of shared governance among clinical nurses. **Methods:** This quantitative study utilized a cross-sectional design and was carried out in three governmental hospitals in the KSA. The sample comprised 669 nurses, who were selected using a convenience sampling method. The Index of Professional Nursing Governance (IPNG) tool was utilized for data collection conducted between February 2025 and April 2025. Descriptive statistics alongside the *t*-test and analysis of variance (ANOVA), Pearson-r correlation coefficient, and multiple linear regression were utilized for data analysis. Significant findings were drawn when *p* ≤ 0.05. **Results:** The average perception of shared governance among clinical nurses at the UBC level was 180.42 out of 430, suggesting that decision-making occurs collaboratively between nurses and management. Significant differences in the average level of clinical nurses’ perceptions of shared governance were noted in relation to their educational qualifications (*F* = 5.015, *p* = 0.001) and nursing units (*F* = 4.157, *p* = 0.010). The hospital in which clinical nurses were employed (*r* = 0.098, *p* = 0.037) and nursing units (*r* = 0.087, *p* = 0.020) exhibited significant correlations with their overall shared professional governance. Furthermore, the hospital where clinical nurses were employed (*β* = 0.406, *p* = 0.001, 95% confidence interval [CI] = 0.166, 0.646) and nursing units (*β* = 0.326, *p* = 0.038, 95% CI = 0.018, 0.314) served as predictors of their overall professional shared governance. **Conclusions:** Clinical nurses in this study showed an initial or relatively low level of shared governance at the UBC level. The overall finding highlights a critical need for nursing managers and leaders to enhance the level of professional shared governance among clinical nurses, which may result in improved nurse retention and overall quality of nursing care. It is crucial to consider clinical nurses’ educational qualifications and working environment at the UBC level when aiming to enhance their level of professional shared governance.

## 1. Introduction

The fundamental principle of professional governance as a form of organizational innovation is to empower nurses with legitimate shared decision-making authority regarding their nursing practice [1,2]. This empowerment ensures that nurses are responsible, accountable, and possess ownership over decisions pertaining to their professional nursing practice [2,3,4]. In alignment with nurses’ empowerment, recent studies have demonstrated that the professional practices of nurses are enhanced and aligned when shared governance frameworks are implemented within organizations [5,6,7]. By engaging in and leveraging shared governance, organizations can connect with a greater number of nurses, leading to more significant outcomes compared with operating in isolation [8,9].

In the context of shared governance, the active involvement of nurses in decision-making processes plays a crucial role in shaping nursing practice [10]. Although numerous healthcare organizations have implemented shared governance within their frameworks, this practice does not inherently ensure the integration of the fundamental principles of shared governance [11]. Globally, various studies have indicated that nurses’ perceptions of professional governance are either suboptimal or fall below the expected level of shared governance within organizations [12,13,14], including the Kingdom of Saudi Arabia (KSA) [7,15,16]. Moreover, the literature indicates that the professional shared governance among nurses can vary significantly between various organizations [7,14,17], and across different nursing units or at unit-based council (UBC) level [15,18,19,20].

In the KSA, a recent study highlights a lack of thorough investigation into shared governance, suggesting that nurses require substantial awareness of this concept [21], particularly at the UBC level. Similarly, another study found that a majority of nurses reported reliance on traditional decision-making processes, noting a deficiency in shared governance structures within their organizations [22]. Given this evidence, the existing deficiency in awareness and dependence on traditional governance, along with the presence of ineffective governance frameworks at the UBC level [7,15,16,21,22], necessitates further investigation. Hence, the aim of the current study was to investigate clinical nurses’ self-assessed perceptions of shared governance at the UBC level within hospital institutions in the KSA, as well as the variations and correlations in their perceptions. In addition, it aimed to analyze the demographic and work-related factors affecting clinical nurses’ perceptions of shared governance at the UBC level.

## 2. Materials and Methods

### 2.1. Design, Sample, and Setting

This quantitative study, utilizing a cross-sectional, correlational design, was conducted across three hospitals within a tertiary, academic medical city located in Riyadh, KSA. The study population comprised clinical nurses employed at the aforementioned hospitals, encompassing various nursing units.

Sample size was determined by utilizing Epi Info software version 7, in consultation with a university statistician, with a confidence interval of 95% and a power of 0.80, based on the total number of clinical nurses working in the three hospitals. Following the calculations, a questionnaire (i.e., online survey) was disseminated to 680 participants, 669 (98.3%) of whom responded to the survey. Convenience sampling was utilized to select participants from all departments within the three hospitals, ensuring they met the established inclusion criteria.

### 2.2. Inclusion and Exclusion Criteria

Registered nurses working at the nursing UBC level with at least six months of employment in any of the three hospitals, and who provided informed consent, were included in the survey. Chief nursing officers, nursing directors, nurse service managers, nurse supervisors, nurse managers, nursing aides, and healthcare assistants were excluded from participating in the survey.

### 2.3. Study Instrument

The demographic characteristics of clinical nurses were assessed based on the following variables: age, gender, educational attainment, job position, years of experience, the hospital where they were employed, and nursing units.

For the purposes of the present study, the Index of Professional Nursing Governance (IPNG), developed by Hess [23], was utilized. IPNG comprises 86 questions designed to assess healthcare personnel’s perceptions of governance along a spectrum ranging from traditional governance to shared governance and ultimately to self-governance. The scoring for the questionnaire was based on a 5-point Likert scale, in which 1 indicates nursing management/administration only, 2 signifies primarily nursing management/administration with some input from clinical nurses, 3 represents an equal sharing of governance responsibilities between clinical nurses and nursing management/administration, 4 denotes primarily clinical nurses with some involvement from nursing management/administration, and 5 reflects clinical nurses exclusively. IPNG encompasses six dimensions: (a) “Nursing Personnel,” which consists of 22 items assessing who controls nursing personnel and associated structures; (b) “Information,” which includes 15 items concerning access to information pertinent to governance activities; (c) “Resources,” comprising 13 items that evaluate who influences practice; (d) “Participation,” which contains 12 items related to involvement in governance structures at various organizational levels; (e) “Practice,” consisting of 16 items that measure control over professional practice; and (f) “Goals,” which includes 8 items that pertain to who establishes and negotiates conflict resolution at different organizational levels.

In the interpretation of the IPNG scoring system, Likert scores of 1 and 2 signified a “decision-making process primarily influenced by management/administration.” Scores exceeding 3 suggested “increased involvement of clinical nurses in decision-making processes.” A relatively high score indicated that clinical nurses are actively engaged in decision-making process. The IPNG total score range, indicative of a traditional (management) decision-making environment, spans from 86 to 172. Conversely, a hospital environment employing shared decision-making between clinical nurses and management would exhibit an IPNG range of 173 to 344. If clinical nurses constitute the decision-making group, then the IPNG range would extend from 345 to 430. To facilitate the interpretation of the results, the total scores were calculated for each subscale by dividing by the number of items within that subscale. Consequently, for all subscales (or the total score), a score exceeding two but falling below four would indicate a certain degree of shared governance [24].

In the current study, a pilot test was carried out involving 35 registered nurses who were not included in the main study. The aim of the pilot study was to evaluate the reliability of the IPNG tool by examining its internal consistency. The findings from the reliability tests indicated acceptable Cronbach’s alpha values for the overall IPNG tool (0.95) as well as for its six dimensions: nursing personnel (0.91), information (0.89), resources (0.88), participation (0.92), practice (0.93), and goals (0.88).

### 2.4. Ethical Considerations and Data Collection

The current study received ethical clearance from the ethics committee of the university medical city (reference number: E-25-9578). Participants were provided with comprehensive information on the aim and objectives of the study, highlighting the voluntary nature of their involvement. Furthermore, clinical nurses were assured that confidentiality of their participation would be strictly maintained, and their responses would be treated completely anonymously. Informed consent was secured before starting to answer the online survey through Google Form. In particular, clinical nurses were asked to respond to the following statement before starting the survey: ‘If you are willing to participate in this online survey, please click “Next” to begin.’

After obtaining ethical approval, data collection was conducted through the administration of an online survey directed at clinical nurses. The researcher asked for administrative approval and requested assistance from the office of the chief nursing officer through the research unit of the university medical city to disseminate the link to the online survey. Specifically, the link to the survey was distributed electronically via institutional emails of clinical nurses and shared in their WhatsApp group chat at the UBC to improve accessibility and completion rates. Furthermore, a poster with a QR code for the online survey was displayed on the bulletin boards of the nursing units across the three hospitals. The researcher assumed responsibility for reviewing the completed forms and consistently monitoring the Google Form for updates on responses. The survey was structured to integrate with Google Sheets, ensuring that the responses from the clinical nurses remained completely anonymous (e.g., no identifying information, such as email address, was collected) and reducing the need for manual data entry. Data were collected from February 2025 to April 2025.

### 2.5. Data Analysis

Data analysis was performed utilizing the Statistical Package for Social Sciences (SPSS) version 28 (International Business Machines [IBM], Armonk, NY, United States of America [USA]). Data extraction from the Google Form through an Excel document was executed, followed by the appropriate method of data entry, data cleaning, coding, and the creation of frequency tables for all study variables. Descriptive statistics, including mean, standard deviation, and frequency, were utilized to summarize the perceptions of shared professional governance among clinical nurses and their demographic characteristics. Normality tests were performed to identify an appropriate test for examining the differences and relationships between sociodemographic variables and perceived professional governance of the clinical nurses. Assessment of the normality of data distribution employed both tests of Kolmogorov–Smirnov (*p* = 0.08) and Shapiro–Wilk (*p* = 0.22). The results from these normality tests (*p* > 0.05) indicated that the data conformed to a normal distribution, which justified the utilization of parametric inferential tests.

The statistical tests included the independent sample *t*-test to assess the difference in the mean of the dependent variable (clinical nurses’ shared governance) concerning dichotomous categorical independent variables, and one-way analysis of variance (ANOVA) to evaluate the difference in the mean of the dependent variable in relation to categorical independent variables with over two categories. Pearson-r correlation test was performed to determine the relationships, while multiple linear regression analysis was conducted to determine the predictors of shared professional governance among clinical nurses. Significant results were inferred if *p* ≤ 0.05.

## 3. Results

### 3.1. Demographic and Work-Related Characteristics of Clinical Nurses

Out of 680 clinical nurses who were invited to participate in the online survey, 669 responses were included in the final statistical analysis, resulting in a response rate of 98.3%. Table 1 presents the demographic and work-related characteristics of clinical nurses. Majority of the sample (90.1%) comprised females, while 9.9% were males. The highest proportion of clinical nurses was within the age range of 30 to 39 years (44.1%), held a baccalaureate degree in nursing (58.0%), employed as staff nurses (96.7%), had 6 to 10 years of experience (46.0%), employed at Hospital A (55.8%), and assigned in the medical unit (15.3%).

### 3.2. Clinical Nurses’ Perceptions on Professional Shared Governance

For the perceptions of nurses on shared professional governance (see Table 2), the domain with the highest mean perception score was identified as “Negotiation and resolution of conflict,” which achieved a mean score of 19.25 out of 40. By contrast, the domain with the lowest mean score was “Control professional practice,” which recorded a mean score of 29.1 out of 80. Overall, the total mean score reflecting clinical nurses’ perceptions of shared governance was 180.42 out of 430, suggesting that shared decision-making occurs between nurses and management at the UBC level.

### 3.3. Differences in Clinical Nurses’ Perceptions of Professional Shared Governance

The findings of this study indicate a notable disparity in the average perception of shared governance among nurses based on their educational attainment (*F* = 5.015, *p* < 0.001) and nursing units (*F* = 4.157, *p* = 0.010). The Tukey post hoc analysis demonstrated that this difference exists between clinical nurses with a bachelor’s degree and those holding a diploma, with the latter group exhibiting a significantly higher perception of shared governance compared with their bachelor’s degree holder counterparts. Furthermore, variations existed between the medical unit and several other units, where clinical nurses assigned to the medical unit demonstrated significantly higher perceptions in comparison to their colleagues assigned to other units (Table 3).

Conversely, the analysis revealed no significant differences in the average perception of shared governance among nurses when considering their age, years of experience, and job position (*p* > 0.05). Furthermore, there were no significant differences in the average perception of shared governance in relation to gender and work environment or hospital (*p* > 0.05), except in nursing units or at the UBC level (*p* = 0.010).

### 3.4. Results of Pearson-r Correlation and Multiple Linear Regression Analysis

Table 4 presents the study findings related to test of relationships and multiple linear regression analysis. Significant correlations were established between the hospital in which clinical nurses were employed (*r* = 0.098, *p* = 0.037) and nursing units (*r* = 0.087, *p* = 0.020), and the overall professional shared governance of clinical nurses.

In addition, the regression model for the overall shared professional governance yielded statistically significant results (*F* [7, 316] = 2.876, *p* = 0.004), explaining approximately 26.8% of the variance (*R*^2^ = 0.065, Adjusted *R*^2^ = 0.043). The regression results revealed that hospital, as a working environment of clinical nurses (*β* = 0.406, *p* = 0.001, 95% confidence interval [CI] = 0.166, 0.646) and nursing units (*β* = 0.326, *p* = 0.038, 95% CI = 0.018, 0.314) were identified as predictors of their overall professional shared governance.

## 4. Discussion

This study addresses a critical issue in nursing practice, which is shared governance. The phenomenon of shared governance among clinical nurses in the KSA has not been extensively examined. Shared governance is established as a management innovation aimed at enhancing the quality of patient care, nursing job satisfaction, empowerment, productivity, and nurse retention [7,25]. The implementation of nursing shared governance practices and organizational frameworks has been widely advocated as an effective approach to improving the work environment by empowering clinical nurses to take responsibility for defining and regulating nursing practice; this approach grants nurses control over their practice and enables them to extend their influence into administrative domains previously managed solely by administrators [15,25].

In the present study, a certain degree of shared governance has been observed among clinical nurses in their workplace, reflected in a mean score of 180.42, suggesting that decisions are collaboratively made between clinical nurses and management. This finding implies that the nurses involved in this study perceive they possess control and some degree of authority over various departmental procedures, including the assignment of daily patient care, processes of patient admissions and discharges, procurement and monitoring of supplies, and creation of work schedules. This result is in contrast with Ghanem Atalla et al. [26], who revealed that Egyptian nurses had a lower overall shared governance score of 109.18. This finding is also consistent with an earlier study in the KSA by Speroni et al. [7], who reported total scores of 106.7 for Magnet hospitals and 101.3 for non-Magnet hospitals. Conversely, a different study in the KSA indicated a relatively high total score of 196.9 among nurses employed at one of the tertiary hospitals affiliated with the Saudi Ministry of Health [16]. Furthermore, the findings of the present study align with those of Butts et al. [27] and Afeef et al. [28], demonstrating that decision-making is shared equally between nurses and management, with a mean score of 32 for this subscale, consistent with the current study. However, George et al. [29] indicated that nurses’ perceptions of their work environment are closely aligned with a traditional governance framework. Aiken et al. [30] found that nurses feel they have limited opportunities to engage in committees related to multidisciplinary professionalism, organizational budgets and expenses, staff scheduling, and strategic planning.

For the relationship between shared governance among clinical nurses and their demographic characteristics, the current study found no significant correlations, except for the hospital where clinical nurses were employed and nursing units. These results diverge from the findings of Kamel and Mohammed [31], who reported a significant association between nurses’ perceptions of shared governance and their job titles. For the variations in nurses’ perceptions of shared governance in relation to their experience, the current study’s findings do not align with Al-Faouri et al. [18], who revealed a significant association between nurses’ perceptions of shared governance and their levels of experience. Conversely, the current study’s results concerning nurses’ gender are consistent with Al-Faouri et al. [18], who found no significant association between nurses’ perceptions of shared governance and their gender. Nevertheless, these findings of the present study are consistent with those conducted in Egypt [26] and Canada [32].

The findings of the present study also align with a previous study in the USA [33], which indicated no significant correlation between gender and clinical nurses’ perceptions of shared governance. Conversely, the results of this study diverge from a previous study of nurses in Egypt [26], demonstrating a significant association between gender and the average score of shared governance among nurses. For the working environment (hospital) and nursing units of the study participants and the average score of shared governance, the significant associations may be attributed to the heightened initiatives aimed at enhancing the implementation of shared governance principles in Saudi hospitals. Many hospitals in the KSA are currently pursuing accreditation, not only at the local level through the Central Board for Accreditation of Healthcare Institutions (CBAHI) [34], but also through Magnet Recognition [35]. In the pursuit of these accreditations, the implementation of shared governance within healthcare organizations is being emphasized and promoted [34]. However, other study variables such as age, gender, educational attainment, job position, and years of experience had no significant relationships with the overall shared governance. These findings could be generally the result of the traditional nursing practice framework in the KSA, which does not permit nurses to operate independently. Nevertheless, these findings should be interpreted with caution, because numerous hospital institutions within the kingdom, similar to the three hospitals within the current study setting [35], apart from the sole two Magnet-recognized hospitals since 2013 [36], are pursuing Magnet recognition. This undertaking promotes shared governance within the organization, thereby possibly influencing the perceptions of clinical nurses.

The results of the current study also revealed that two work-related factors, namely, the hospital as the working environment of clinical nurses and nursing units, significantly predicted the overall shared professional governance. These findings align with an earlier study conducted in the USA, which determined that a professional working environment integrating shared governance could act as a valuable intervention for hospital organizations, thereby facilitating optimal outcomes for patients and clinical nurses [20]. This assertion is further corroborated by the American Nurses Association, which suggests that shared governance among clinical nurses acts as a structural framework that enables them to articulate and oversee their practice with professional accountability, autonomy, equity, partnership, and ownership, thereby empowering them with decision-making authority [37]. This finding may imply that hospital institutions upholding shared governance among clinical nurses, such as the present study’s working environment during its pursuit of Magnet accreditation, provide them with a platform to express their opinions and opportunities for involvement in decision-making process within the organization. Thus, clinical nurses are able to participate in organizational decision-making to fulfill their professional obligations.

### Limitations of the Study

This study presents several limitations warranting careful consideration. The reliance on self-reported shared governance, coupled with the use of a convenience sampling approach, may introduce a degree of bias in the responses of clinical nurses participating in the survey. It could be possible that clinical nurses with strong negative views on shared governance, or alternatively, those who held very positive opinions, were more inclined to participate in the study. Consequently, the findings should be approached with caution, given that the ratings provided by clinical nurses may not truly represent the shared governance among nurses in various healthcare settings throughout the KSA, including private hospitals. Furthermore, the notion of shared governance, as part of the ongoing effort to achieve Magnet Recognition, may hinder participants from fully grasping the significance of shared governance in nursing practice, thereby possibly influencing the outcomes. Lastly, the limited number of management-level respondents who participated in the current study is another recognized limitation due to the disproportion in the sample. Specifically, out of the 669 survey respondents, majority were staff nurses, while only a small number were management-level respondents (Charge nurse/Team leader = 22, 3.3%). This disparity restricted the study’s capacity to thoroughly assess the comparison of perceived shared governance due to the insufficient representation of charge nurses or team leaders. Caution must be exercised when extrapolating findings to management-level respondents (charge nurses or team leaders), given their limited number in the sample. Therefore, the sample may not accurately reflect the management-level nurses within the context of the study. Future research should aim for more stratified sampling to ensure that essential subgroups (e.g., charge nurses or team leaders, nurse managers, and other higher management-level nurses) are sufficiently represented, along with a larger sample size across various regions in the KSA. Therefore, these constraints limit the generalizability of this study’s findings.

## 5. Conclusions

Overall, the clinical nurses in this study exhibited an initial or relatively low level of shared governance consistent with previous studies in the KSA and globally. The findings suggest that the perception of nurses concerning shared governance at the UBC level was found to be unsatisfactory and varied, highlighting a significant necessity for nursing managers and leaders in the KSA to improve the level of shared governance among clinical nurses.

## 6. Implications for Clinical Nursing Practice

The findings of the present study can be leveraged to strengthen the shared governance framework between clinical nurses and healthcare institutions in the KSA, including those processing accreditations locally or internationally, such as Magnet Recognition. This approach can foster a markedly conducive work environment for clinical nurses, ultimately leading to improved quality nursing care. In addition, this method empowers nurse managers, thereby enhancing their influence in leadership and management roles.

Considering clinical nurses’ educational qualifications and their hospital environment at the UBC level, it is crucial to develop and provide educational training and supportive strategies for these nurses. Such strategies should focus on promoting and establishing shared governance in the practice of clinical nurses. Lastly, the adoption of shared governance in the nursing practice must be promoted to elevate the authority and influence of clinical nurses within healthcare environments.

## Figures and Tables

**Table 1 nursrep-15-00426-t001:** Demographic and work-related characteristics of clinical nurses (N = 669).

Demographic Characteristics	f	%
Gender		
Male	66	9.9
Female	603	90.1
Age		
<30 years	204	30.5
30–39 years	295	44.1
40–49 years	127	19.0
>50 years	43	6.4
Educational attainment		
Nursing diploma	249	34.5
Bachelor’s degree in nursing	388	58.0
Master’s degree in nursing	32	4.8
Job position		
Staff nurse	647	96.7
Charge nurse/Team leader	22	3.3
Years of experience		
<5 years	90	13.5
6–10 years	308	46.0
11–15 years	94	14.1
>15 years	177	26.5
Hospital		
Hospital A	373	55.8
Hospital B	194	29.0
Hospital C	102	15.2
Nursing units (UBC)		
Medical	102	15.3
Surgical	83	12.4
Critical/Intensive care	57	8.5
Operating room	66	9.9
Recovery room	51	7.6
Emergency department	55	8.2
Outpatient department	58	8.7
Maternity/delivery	53	7.9
Pediatric	57	8.5
Psychiatric	21	3.1
Rehabilitation	18	2.7
Oncology	26	3.9
Nursing education	22	3.3

Note. UBC = Unit-based Council; f = Frequency; % = Percentage.

**Table 2 nursrep-15-00426-t002:** Clinical nurses’ perceptions on professional shared governance (N = 669).

Domains	Range	Mean	SD
Control over personnel	22–110	45.78	18.49
Access to information	15–75	33.25	12.33
Influence over resources	13–65	27.17	11.11
Participation in committee structures	12–60	23.78	9.40
Control over professional practice	16–80	29.01	11.41
Negotiation and conflict resolution	8–40	19.25	8.82
Total shared governance	86–430	180.42	59.18

Note. SD = Standard Deviation.

**Table 3 nursrep-15-00426-t003:** Differences in clinical nurses’ perceptions of professional shared governance.

Variables	N	Mean	SD	Computed Value	*p*
Age					
<30 years	204	174.34	51.27	*F* = 1.657	0.175
30–39 years	295	185.57	65.25		
40–49 years	127	180.50	57.33		
>49 years	43	173.48	53.64		
Years of experience					
≤5 years	90	178.46	53.82	*F* = 1.311	0.270
6–10 years	308	178.64	60.07		
11–15 years	94	191.63	68.07		
>15 years	177	178.54	54.84		
Educational attainment					
Nursing diploma	249	189.81	68.21	*F* = 5.015	0.001 **
Bachelor’s degree in nursing	388	173.66	51.59		
Master’s degree in nursing	32	178.00	46.86		
Gender					
Male	66	185.28	70.80	*t* = 0.703	0.483
Female	603	179.89	57.80		
Job position					
Clinical staff nurse	645	180.58	59.28	*t* = 0.090	0.914
Charge nurse/Team leader	22	177.12	66.91		
Hospital					
Hospital A	373	188.45	77.12	*F* = 1.822	0.317
Hospital B	194	172.68	66.25		
Hospital C	102	177.07	71.64		
Nursing units (UBC)					
Medical	102	186.12	78.45	*F* = 4.157	0.010 *
Surgical	83	181.25	60.23		
Critical/Intensive care	57	175.32	67.15		
Operating room	66	177.54	62.45		
Recovery room	51	173.56	68.55		
Emergency department	55	184.23	71.12		
Outpatient department	58	183.89	69.15		
Maternity/Delivery	53	188.91	67.88		
Pediatric	57	175.84	59.51		
Psychiatric	21	169.28	56.45		
Rehabilitation	18	168.59	60.27		
Oncology	26	172.17	65.48		
Nursing education	22	190.46	77.69		

Note. UBC = Unit-based council; *F* = value in analysis of variance; *t* = value in *t*-test; SD = Standard Deviation; *p* = *p*-value. * Significance level at *p* ≤ 0.01; ** Significance level at *p* ≤ 0.001.

**Table 4 nursrep-15-00426-t004:** Results of Pearson-r Correlation and Multiple Linear Regression Analysis (*n* = 669).

Study Variables	*r*	*p*	Unstandardized Coefficients		Standardized Coefficients	*t*	*p*	95% Confidence Interval	
			*β*	*SE-b*	*β*			Lower Limit	Upper Limit
Age	0.056	0.340	0.098	0.087	0.062	1.122	0.263	0.074	0.270
Gender	−0.015	0.312	−0.121	0.099	−0.069	−1.226	0.221	−0.316	0.073
Educational attainment	−0.176	0.355	−0.004	0.010	−0.038	−0.414	0.679	−0.024	0.016
Job position	−0.026	0.078	−0.013	0.061	−0.019	−0.208	0.136	−0.132	0.107
Years of experience	−0.125	0.287	−0.032	0.053	−0.040	−0.604	0.546	−0.136	0.072
Hospital	0.098	0.037 *	0.406	0.122	0.206	3.327	0.001 **	0.166	0.646
Nursing units (UBC)	0.087	0.020 *	0.326	0.095	0.125	0.248	0.038 *	0.018	0.314
R^2^ (Adjusted R^2^) = 0.065 (0.043); *F* [7, 316] = 2.876; *p* = 0.004									

Note. Total professional shared governance is the dependent variable. UBC = Unit-based council; *r* is the Pearson-r correlation value; *p* is the *p*-value; *β* is the unstandardized coefficient; *SE-b* is the standard error; *t* is the *t*-value. * Significance level at *p* ≤ 0.05; ** Significance level at *p* ≤ 0.001.

## Data Availability

Data is contained within the article.

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
