# Peer review of "Clinical Nurses’ Involvement in Decision-Making Process at the Nursing Unit-Based Council Level: A Cross-Sectional Study of Shared Professional Governance in the Kingdom of Saudi Arabia"

_nursrep, 2025, doi:10.3390/nursrep15120426_

Round 1

Reviewer 1 Report

Comments and Suggestions for Authors

Dear Authors, your manuscript addresses an important topic on shared governance and nurses’ participation in decision-making within Saudi Arabia. Minor revisions are recommended:

Abstract: Grammar should be checked. Repetition should be taken into account. Furthermore, the findings section of the abstract should be strengthened with key statistical information.
Introduction: Many of the sentences are very long, and a lot of information is contained within a single sentence. Furthermore, some phrases are repeated frequently, making reading difficult. Therefore, I recommend revising the entire article accordingly.
Method: It is written in a very detailed and scientifically sound manner. However, I recommend revising this section due to the long sentences and frequent repetitions in other sections. Furthermore, the scale used is given in more detail, and a more concise form is possible. IPNG's Cronbach's alpha is not specified in this much detail. You mentioned the use of inclusion criteria; however, these criteria were not explicitly defined. A flow diagram was included to illustrate the recruitment and inclusion process of the participants, showing the number of nurses invited, excluded, and included in the final analysis. Findings: The sentences "Majority of the sample (90.1%) comprised females, whereas 9.9% were males. The sample indicated that 67.9% were females and 32.1% were males" should be checked. Additionally, the R2 expression in the logistic regression should be checked for consistency within the text. The Discussion section comprehensively compares the study's findings with previous literature. However, long sentences and repetitions should also be reviewed in this section.

Author Response

REVIEWER #1 COMMENTSs

Dear Authors, your manuscript addresses an important topic on shared governance and nurses’ participation in decision-making within Saudi Arabia. 

AUTHOR’S REPLY: Dear Honorable Reviewer #1, thank you very much for your valuable feedback. My point-by-point response to each of your review comments is indicated below. Correspondingly, I specified the location of the required revisions in the revised version of my work.

Minor revisions are recommended:

Abstract: Grammar should be checked. Repetition should be taken into account. Furthermore, the findings section of the abstract should be strengthened with key statistical information.

AUTHOR’S REPLY: The revised version of my work has been submitted for second round of English language editing. Repetitive entries have been corrected, and key statistical results have been added. Please refer to Line/s 10-46.

Introduction: Many of the sentences are very long, and a lot of information is contained within a single sentence. Furthermore, some phrases are repeated frequently, making reading difficult. Therefore, I recommend revising the entire article accordingly.

AUTHOR’S REPLY: Long sentences have been shortened and repeated phrases have been deleted. Please refer to Line/s 50-95. These revisions apply to similar instances throughout the revised version of my work.

Method: It is written in a very detailed and scientifically sound manner. However, I recommend revising this section due to the long sentences and frequent repetitions in other sections. Furthermore, the scale used is given in more detail, and a more concise form is possible. IPNG's Cronbach's alpha is not specified in this much detail.

AUTHOR’S REPLY: This has been addressed throughout the Methods section. IPNG and its subscales’ Cronbach's alpha values have been added in Line/s 156-161.

You mentioned the use of inclusion criteria; however, these criteria were not explicitly defined.

AUTHOR’S REPLY: The inclusion and exclusion criteria have been added in Line/s 103-107.

A flow diagram was included to illustrate the recruitment and inclusion process of the participants, showing the number of nurses invited, excluded, and included in the final analysis.

AUTHOR’S REPLY: This has been presented in narrative format in Line/s 108-114 and 213-215.

Findings: The sentences "Majority of the sample (90.1%) comprised females, whereas 9.9% were males. The sample indicated that 67.9% were females and 32.1% were males" should be checked.

AUTHOR’S REPLY: Thank you for observing this and I sincerely apologize for this typo error. This has been addressed in Line/s 215-229.

Additionally, the R2 expression in the logistic regression should be checked for consistency within the text.

AUTHOR’S REPLY: The R2 expression has been added at the end part of Table 4. This has been addressed in Line/s 268-281 and Table 4. I kindly inform the Honorable Reviewer #1 that ‘nursing units’ has been added in Tables 1, 3 and 4 as well as in the main text, as suggested by the Esteemed Reviewer #2.

The Discussion section comprehensively compares the study's findings with previous literature. However, long sentences and repetitions should also be reviewed in this section.

AUTHOR’S REPLY: Long sentences have been broken down into shorter, concise and more readable ones to properly and clearly emphasize the study’s flow of ideas. This revision applies to similar instances throughout the revised version of my work. Additionally, the revised version has undergone second round of English language editing before resubmission.

Reviewer 2 Report

Comments and Suggestions for Authors

This study adopted a cross-sectional research design to focus on the level of awareness of shared professional governance among clinical nurses in Saudi Arabia, as well as the influence of demographic and work-related factors on their awareness. The topic has practical value, the sample size is sufficient, and the results can provide a reference for improving the shared professional governance in nursing in Saudi Arabia. The following aspects can be improved:

  1. The research background does not clearly indicate "research gaps" or "innovative points". It is recommended to provide such information.
  2. In the purpose section of the abstract, only the statement "analyzing the influence of demographic and job-related factors" is given, without explicitly mentioning the core research scenario of "UBC", which is inconsistent with the title of the article.
  3. In Section 2 "Design, Sample and Settings", it is not specified what these "established inclusion criteria" are, and no exclusion criteria are mentioned at all. It is recommended to provide such information.
  4. In Section 2 "Research Tools", the IPNG scale was not translated into the Arabic language, and the results of the local version's reliability and validity tests (such as Cronbach's α and content validity index) were not reported.
  5. According to the scoring criteria of the IPNG scale, a score of 180.42 is only slightly above the lower limit of the scope of shared governance. This should rather be interpreted as "at the initial or relatively low level of shared governance". The current description weakens the core finding of "still requiring significant improvement" and contradicts the conclusion in the following text.
  6. In the results section of the third section, when referring to the sample size, the statement "90.1% are female" contradicts the following sentence "67.9% are female and 32.1% are male". It is recommended that this be verified and corrected.
  7. This study employed convenience sampling, with the sample concentrated in the same medical complex. 96.7% of the participants were regular nurses, while only 2.4% held positions as head nurses or supervisory nurses. The sample size for the management level was small, and there was a selection bias.
  8. In the conclusion of Section 5, it was stated that "a shared governance model needs to be developed to increase the retention rate of nurses and improve the quality of nursing care". However, no core elements of the model were proposed based on the results of this study, and the conclusion lacks operability.
  9. References 4 and 5 are cited repeatedly; the formats of some references are inconsistent and do not conform to the citation norms of academic journals. It is recommended to make the necessary revisions.

Author Response

REVIEWER #2 COMMENTS

This study adopted a cross-sectional research design to focus on the level of awareness of shared professional governance among clinical nurses in Saudi Arabia, as well as the influence of demographic and work-related factors on their awareness. The topic has practical value, the sample size is sufficient, and the results can provide a reference for improving the shared professional governance in nursing in Saudi Arabia.

AUTHOR’S REPLY: Dear Esteemed Reviewer #2, thank you so much for your valuable feedback that led to the improvement of the revised version of my paper. My point-by-point response to each of your review comments is indicated below. Correspondingly, I specified the location of the required revisions in the revised version of my work.

The following aspects can be improved:

  1. The research background does not clearly indicate "research gaps" or "innovative points". It is recommended to provide such information.

AUTHOR’S REPLY: This has been addressed in the Abstract (Line/s 10-17) as well as in the Introduction section in Line/s 84-95.

  1. In the purpose section of the abstract, only the statement "analyzing the influence of demographic and job-related factors" is given, without explicitly mentioning the core research scenario of "UBC", which is inconsistent with the title of the article.

AUTHOR’S REPLY: This has been addressed in Line/s 18-19 and throughout the Abstract and main text. This resulted in adding nursing units’ in Tables 1, 3 and 4 as well as in the main text, as suggested.

  1. In Section 2 "Design, Sample and Settings", it is not specified what these "established inclusion criteria" are, and no exclusion criteria are mentioned at all. It is recommended to provide such information.

AUTHOR’S REPLY: The inclusion and exclusion criteria have been added in Line/s 103-107.

  1. In Section 2 "Research Tools", the IPNG scale was not translated into the Arabic language, and the results of the local version's reliability and validity tests (such as Cronbach's α and content validity index) were not reported.

AUTHOR’S REPLY: This has been addressed throughout the Methods section. IPNG and its subscales’ Cronbach's alpha values have been added in Line/s 156-161.

  1. According to the scoring criteria of the IPNG scale, a score of 180.42 is only slightly above the lower limit of the scope of shared governance. This should rather be interpreted as "at the initial or relatively low level of shared governance". The current description weakens the core finding of "still requiring significant improvement" and contradicts the conclusion in the following text.

AUTHOR’S REPLY: This has been addressed, and the term ‘suboptimal’ has been changed to ‘relatively low’ in the conclusion of Abstract section and main text in Line/s 39 and 397, respectively.

  1. In the results section of the third section, when referring to the sample size, the statement "90.1% are female" contradicts the following sentence "67.9% are female and 32.1% are male". It is recommended that this be verified and corrected.

AUTHOR’S REPLY: Thank you for observing this and I sincerely apologize for this typo error. This has been addressed, verified and corrected in Line/s 215-229.

  1. This study employed convenience sampling, with the sample concentrated in the same medical complex. 96.7% of the participants were regular nurses, while only 2.4% held positions as head nurses or supervisory nurses. The sample size for the management level was small, and there was a selection bias.

AUTHOR’S REPLY: This has been addressed in a separate sub-section of Limitations of the Study in Line/s 383-395.

  1. In the conclusion of Section 5, it was stated that "a shared governance model needs to be developed to increase the retention rate of nurses and improve the quality of nursing care". However, no core elements of the model were proposed based on the results of this study, and the conclusion lacks operability.

AUTHOR’S REPLY: Thank you for observing this. This has been revised in the conclusion of Abstract section and main text in Line/s 43-46 and 402-404, respectively.

  1. References 4 and 5 are cited repeatedly; the formats of some references are inconsistent and do not conform to the citation norms of academic journals. It is recommended to make the necessary revisions.

AUTHOR’S REPLY: This has been addressed in the References section. Please be assured that the revised version of my work has now fully adhered to Nursing Reports’ citation and referencing style. Please refer to Line/s 435-520.

Reviewer 3 Report

Comments and Suggestions for Authors

This study investigates clinical nurses’ participation in shared governance in Saudi Arabia, an important and timely topic in nursing leadership and organizational development. The manuscript is well structured and presents valuable findings. However, several sections could be improved to enhance clarity and academic rigor:

Introduction:

The background is comprehensive but too general. Please focus on the research gap and the specific rationale for exploring shared governance at the unit-based council level.

Remove redundant citations and ensure each reference directly supports the argument.

Methods:

Clarify inclusion/exclusion criteria and the justification for using convenience sampling.

Include reliability statistics (e.g., Cronbach’s alpha) for the IPNG instrument in this sample.

State the ethical approval number in-text.

Results:

Review numerical consistency (e.g., gender percentages and educational categories).

Explain the discrepancy between the stated explained variance (23.3%) and the reported R² (0.054).

Provide a brief interpretation of the regression coefficients.

Discussion and Conclusions:

Deepen the discussion by linking findings to the cultural and organizational context of Saudi Arabia.

Highlight the practical implications for nurse managers and health policymakers.

Avoid excessive repetition of descriptive findings.

Tables and Figures:

Generally clear, but ensure consistent formatting, proper labeling, and alignment with journal style.

Language and Style:

The English is understandable but requires moderate editing for conciseness, grammar, and flow.

Shorten long sentences and remove redundancies between sections.

Overall, the manuscript makes a meaningful contribution to nursing management literature. After improving methodological transparency and language clarity, it will be well-suited for publication.

Author Response

REVIEWER #3 COMMENTS

This study investigates clinical nurses’ participation in shared governance in Saudi Arabia, an important and timely topic in nursing leadership and organizational development. The manuscript is well structured and presents valuable findings.

AUTHOR’S REPLY: Dear Respected Reviewer #3, please accept my sincere appreciation for your valuable feedback that led to the enhancement of the revised version of my paper. My point-by-point response to each of your review comments is indicated below. Correspondingly, I specified the location of the required revisions in the revised version of my work.

However, several sections could be improved to enhance clarity and academic rigor:

Introduction:

The background is comprehensive but too general. Please focus on the research gap and the specific rationale for exploring shared governance at the unit-based council level.

Remove redundant citations and ensure each reference directly supports the argument.

AUTHOR’S REPLY: This has been addressed in the Abstract (Line/s 10-17) as well as in the Introduction section in Line/s 84-95.

Methods:

Clarify inclusion/exclusion criteria and the justification for using convenience sampling.

AUTHOR’S REPLY: The inclusion and exclusion criteria have been added in Line/s 103-107.

Include reliability statistics (e.g., Cronbach’s alpha) for the IPNG instrument in this sample.

AUTHOR’S REPLY: This has been addressed throughout the Methods section. IPNG and its subscales’ Cronbach's alpha values have been added in Line/s 156-161.

State the ethical approval number in-text.

AUTHOR’S REPLY: This has been added in Line/s 164.

Results:

Review numerical consistency (e.g., gender percentages and educational categories).

Explain the discrepancy between the stated explained variance (23.3%) and the reported R² (0.054).

AUTHOR’S REPLY: These have been addressed in Line/s 212-286. I kindly inform the Respected Reviewer #3 that ‘nursing units’ has been added in Tables 1, 3 and 4 as well as in the main text, as suggested by the Esteemed Reviewer #2.

Provide a brief interpretation of the regression coefficients.

AUTHOR’S REPLY: This has been addressed in Line/s 268-286.

Discussion and Conclusions:

Deepen the discussion by linking findings to the cultural and organizational context of Saudi Arabia.

AUTHOR’S REPLY: This has been addressed comprehensively in Line/s 288-380. Correspondingly, this resulted adding several references.

Highlight the practical implications for nurse managers and health policymakers.

AUTHOR’S REPLY: This has been addressed in the conclusion part of the Abstract in Line/s 38-46 and in the Implications for Clinical Nursing Practice section 405-418.

Avoid excessive repetition of descriptive findings.

AUTHOR’S REPLY: This has been revised in the first part of the Results section Line/s 211-229.

Tables and Figures:

Generally clear, but ensure consistent formatting, proper labeling, and alignment with journal style.

AUTHOR’S REPLY: This has been observed throughout the Results section in Line/s 211-286.

Language and Style:

The English is understandable but requires moderate editing for conciseness, grammar, and flow. Shorten long sentences and remove redundancies between sections.

AUTHOR’S REPLY: Long sentences have been broken down into shorter, concise and more readable ones to properly and clearly emphasize the study’s flow of ideas. This revision applies to similar instances throughout the revised version of my work. Additionally, the revised version has undergone second round of English language editing before resubmission.

Overall, the manuscript makes a meaningful contribution to nursing management literature. After improving methodological transparency and language clarity, it will be well-suited for publication.

AUTHOR’S REPLY: Thank you very much, Respected Reviewer #3. Kindly know that I am open, if necessary, and more than willing to make additional changes for further improvement of my work. More power and Godspeed!